# Transcriptomic Analysis of the Molecular Mechanism Potential of Grafting—Enhancing the Ability of Oriental Melon to Tolerate Low-Nitrogen Stress

**DOI:** 10.3390/ijms25158227

**Published:** 2024-07-27

**Authors:** Yulei Zhu, Ziqing Sun, Hongxi Wu, Caifeng Cui, Sida Meng, Chuanqiang Xu

**Affiliations:** 1College of Horticulture, Shenyang Agricultural University, Shenyang 110866, China; zhuyulei123@stu.syau.edu.cn (Y.Z.); sunziqing2022@stu.syau.edu.cn (Z.S.); wuhongxi2022@stu.syau.edu.cn (H.W.); caifengcui2022@stu.syau.edu.cn (C.C.); 2Key Laboratory of Protected Horticulture (Ministry of Education), Shenyang Agricultural University, Shenyang 110866, China; 3Modern Protected Horticultural Engineering & Technology Center, Shenyang Agricultural University, Shenyang 110866, China; 4Key Laboratory of Horticultural Equipment (Ministry of Agriculture and Rural Affairs), Shenyang Agricultural University, Shenyang 110866, China

**Keywords:** melon, squash, grafting, low-nitrogen stress, RNA-seq

## Abstract

Nitrogen is the primary nutrient for plants. Low nitrogen generally affects plant growth and fruit quality. Melon, as an economic crop, is highly dependent on nitrogen. However, the response mechanism of its self-rooted and grafted seedlings to low-nitrogen stress has not been reported previously. Therefore, in this study, we analyzed the transcriptional differences between self-rooted and grafted seedlings under low-nitrogen stress using fluorescence characterization and RNA-Seq analysis. It was shown that low-nitrogen stress significantly inhibited the fluorescence characteristics of melon self-rooted seedlings. Analysis of differentially expressed genes showed that the synthesis of genes related to hormone signaling, such as auxin and brassinolide, was delayed under low-nitrogen stress. Oxidative stress response, involved in carbon and nitrogen metabolism, and secondary metabolite-related differentially expressed genes (DEGs) were significantly down-regulated. It can be seen that low-nitrogen stress causes changes in many hormonal signals in plants, and grafting can alleviate the damage caused by low-nitrogen stress on plants, ameliorate the adverse effects of nitrogen stress on plants, and help them better cope with environmental stresses.

## 1. Introduction

Nitrogen is an essential component of fertilizers and pesticides and a limiting factor for crop growth [1]. Nitrogen fertilizer is widely used in agriculture to increase crop yields. However, nitrogen deficiency often leads to slow growth and low photosynthetic efficiency, thus affecting plant yield and quality [2,3]. Therefore, plants have evolved a high-transfer operating system versus a low-affinity transfer system for more efficient nitrogen uptake to better respond to changes in the soil environment [4,5]. Leaf nitrogen content correlates positively with plant photosynthesis because about 70% of leaf nitrogen is in chloroplasts [6]. Low-nitrogen stress affects photosynthesis directly or indirectly by influencing various enzymes and fluorescence coefficients in plants. Studies over the past decades have shown that the low-nitrogen stress response is associated with multiple pathways, including signal transduction, oxidative stress, carbon and nitrogen metabolism, and the synthesis of secondary metabolites [7,8]. When plants are subjected to low-nitrogen stress, many stress-related defense response genes are induced to enhance the low nitrogen tolerance of plants. Long-term low nitrogen often can lead to slow development of plants, premature leaf senescence, and ultimately result in declining fruit quality [9]. As we know, phytohormones are microscopic and efficient endogenous substances produced in plants, which play an essential role in plant growth and development, and germination, nutrient growth, reproductive growth, and senescence are accompanied by phytohormone regulation [10]. Various hormones also play a role in plant growth, improving grafting survival, and helping them better cope with environmental stresses [11]. Many studies have shown that plant hormones are important in adaptation to low-nitrogen stress [12]. For example, brassinolide can increase the photosynthetic rate and reduce ROS production, which improves plant growth and development under salt stress [13].

Grafting is an ancient horticultural technology widely used in the cultivation and variety improvement of horticultural crops, creating considerable economic value [14]. Grafting can significantly improve the nitrogen absorption capacity of plants, enhance the absorption and utilization of mineral elements, and ameliorate the adverse effects of nitrogen stress. For example, melon varieties ‘Melina’ and ‘Callicum’ were grafted onto the Indian squash rootstock variety ‘Kamel’ to significantly enhance the low-nitrogen tolerance ability [15]. Under long-term low-nitrogen-stress conditions, the differentially expressed genes (DEG) involved in photosynthesis were almost down-regulated, inhibiting the nitrate metabolic pathways of cucumber and reducing the cucumber plant size and biomass [3]. However, the grafted plants have more substantial nitrogen absorption capacity, and grafted plants were more resistant to low-nitrogen stress [16]. Tomato plants grafted on high-nitrogen-efficient rootstocks can enhance nitrogen absorption and utilization and maintain growth under low-nitrogen stress by increasing the efficiency of light energy utilization [17]. However, the intrinsic molecular mechanisms underlying the strong tolerance of grafted plants to low-nitrogen stress are still unclear.

In this study, we analyzed the transcriptomic data to compare the differences between self-rooted grafted seedlings and squash-rooted seedlings under normal-nitrogen and low-nitrogen conditions, aiming to elucidate the potential molecular mechanisms by which grafting enhances melon tolerance to low-nitrogen stress. We hope our research findings can enrich the theoretical framework of grafting cultivation and provide theoretical references. Additionally, the results may serve as a practical basis for nutrient management of grafted melon cultivation in the greenhouse.

## 2. Results

### 2.1. Effect of Low-Nitrogen Stress on the Fluorescence Characteristics of Self-Rooted Oriental Melon Seedlings

In order to investigate the effect of low-nitrogen stress on the fluorescence characteristics of self-rooted oriental melon seedlings, the quantum yield of unregulated energy dissipation at Y(NO)PSI, the quantum yield of regulated energy dissipation at Y(NPQ)PSII, the qP photochemical quenching coefficient, the photochemical efficiency at Y(I)PSI, and the quantum yield of non-photochemical energy dissipation at Y(NA)PSI due to receptor side limitation were measured under low-nitrogen (LN) and normal-nitrogen (RN) conditions. Compared with RN, the quantum yield of non-regulated energy dissipation at Y(NO)PSI (Figure 1A), the quantum yield of regulated energy dissipation at Y(NPQ)PSII (Figure 1B), the effective photochemical efficiency of Y(I)PSI (Figure 1D), and the quantum yield of non-photochemical energy dissipation at PSI due to the lateral restriction of the Y(NA) acceptor (Figure 1E) were significantly lower after LN treatment, and the qP photochemical quenching coefficient (Figure 1C) was not significantly different. The results indicated that low-nitrogen treatment could significantly reduce the fluorescence characteristics of self-rooted oriental melon seedlings.

### 2.2. Transcriptome Data Analysis and Functional Annotation

In order to reveal the potential molecular mechanism of the response to low-nitrogen stress between the self-rooted oriental melon seedlings and grafted seedlings (oriental melon scion grafted onto squash rootstock), we used RNA-Seq technology to sequence the transcriptomes of leaf samples. A total of 114.82 Gb Clean Data was obtained, with Q30 ≥ 95.04% (sequencing error rate and <0.1%) and GC content >45.39%, indicating that the transcriptomics data met the requirements for subsequent analysis (Table 1). In the principal component analysis (PCA), the principal components (PC1 and PC2) were 69.9% and 18.6%, respectively (Figure 2A), and there was a clear separation of the treatments from the control. The PCA, in combination with the associated heat map, showed good biological reproducibility. Transcriptional trend densitograms of nine samples showed gene expression levers in the range of 10^−2^–10^4^ (Figure 2B). By comparing the self-rooted oriental melon seedlings under normal-nitrogen (RNHBHB) and low-nitrogen treatment (LNHBHB), a total of 1272 differentially expressed genes (DEGs) were identified, of which 761 were down-regulated, which accounted for 59.83% of the total DEGs, and 511 were up-regulated, which accounted for 40.17% of the total genes (Figure 2C). Analyzing the transcriptome data of self-rooted oriental melon seedlings (LNHBHB) and grafted seedlings (LNHBZY65) under low-nitrogen treatment, a total of 1130 DEGs were identified, of which 351 were down-regulated, accounting for 31.06% of the total DEGs, and 779 were up-regulated, accounting for 68.94% of the total DEGs (Figure 2D). The results showed that most DEGs were down-regulated under low-nitrogen stress, and most DEGs were up-regulated by grafting.

### 2.3. GO and KEGG Enrichment Analysis of Differentially Expressed Genes (DEGs)

GO enrichment and DEGs functional classification analysis were performed by comparing RNHBHB with LNHBHB and LNHBHB with LNHBZY65. A total of 1272 genes were annotated in the GO database, which could be categorized into biological pathways (BP), cellular components (CC), and molecular functions (MF) and further subdivided into 53 functional groups. The BP items could be categorized into 20 functional groups, among which “metabolic process”, “cellular process”, and “single-organism process” were significantly enriched. Among the CC categories, “membrane” and “cell” were the most enriched categories. The most abundant of the MF process categories were “binding” and “catalytic activity” (Figure 3A,B). These DEGs labeled in the KEGG database mainly belonged to the metabolic process category. The DEGs were enriched with 20 KEGG pathways compared to the KEGG database. The main KEGG pathways of DEGs enrichment in RNHBHB and LNHBHB were “Plant hormone signal transduction”, “Phenylpropanoid biosynthesis”, “MAPK signaling pathway-plant”, “Pentose and glucuronate interconversions”, “Plant–pathogen interaction”, “Starch and sucrose metabolism” (Figure 3C). The main KEGG pathways of LNHBHB and LNHBZY65 for DEGs enrichment were “Plant hormone signal transduction”, “Starch and sucrose metabolism” and “MAPK signaling pathway-plant”, “Phenylpropanoid biosynthesis”, “Pentose and glucuronate interconversions”, and “Galactose metabolism” (Figure 3D). It could be inferred that the DEGs related to the signal transduction pathway, oxidative stress response, carbon and nitrogen metabolism pathway, and secondary metabolite synthesis had significant changes under low-nitrogen stress.

### 2.4. Differentially Expressed Genes (DEGs) Involved in Signal Transduction

Studies have shown that plant hormones play an important role in plant growth and development and participate in plant stress response [18,19]. We conducted the enrichment analysis of the KEGG pathway to explore further the changes in hormones in self-rooted oriental melon seedlings and grafted seedlings of oriental melon scion grafted onto squash rootstock under low-nitrogen stress. The DEGs were mainly involved in the auxin and brassinolide pathways in the plant hormone signaling pathways. For example, most of the significantly down-regulated DEGs involved in auxin signaling under low-nitrogen stress belonged to the AUX/IAA family, mainly including MELO3C008287.2, MELO3C000885.2, MELO3C005476.2, MELO3C012951.2, and MELO3C027346.2 (Figure 4A). Most of the differentially expressed genes involved in brassinolide signal transduction belonged to TCH4 and CYCD3, which regulate cell elongation and cell division processes. Significantly down-regulated genes included MELO3C002962.2, MELO3C017478.2, MELO3C017480.2, MELO3C002144.2, MELO3C004536.2, MELO3C006805.2, MELO3C014984.2, MELO3C010836.2, MELO3C018476.2, and MELO3C019540.2 (Figure 4B). Grafted seedlings subjected to low-nitrogen-stress treatments compared to self-rooted seedlings, the DEGs (MELO3C008287.2, MELO3C028893.2, MELO3C028973.2, MELO3C013394.2, MELO3C020752.2, MELO3C020771.2, MELO3C034602.2, and MELO3C017825.2) in the auxin signaling pathway were significantly up-regulated, which together regulated cell enlargement and plant growth (Figure 4C). BAK1 (MELO3C014454.2 significantly up-regulated) and BRI1 (MELO3C002144.2, MELO3C002370.2, MELO3C012266.2, MELO3C017651.2, and MELO3C019588.2 significantly up-regulated) of brassinolide signal transduction pathway were activated, which subsequently induced downstream BSK (MELO3C025882.2 significantly up-regulated) changes on CYCD3 (MELO3C010836.2, MELO3C018476.2, MELO3C019540.2 significantly up-regulated) to promote cell division (Figure 4D). In summary, the expression of hormones such as auxin and brassinolide differed between self-rooted and grafted oriental melon seedlings under low-nitrogen stress. Among them, the DEGs were mostly down-regulated compared to RNHBHB and LNHBHB. In contrast, it was significantly up-regulated compared to LNHBHB and LNHBZY65. So, we suggested that plants might improve their low-nitrogen stress tolerance by regulating the rapid synthesis of certain hormones after grafting.

### 2.5. Differentially Expressed Genes (DEGs) Involved in Oxidative Stress

Abiotic stresses can induce the production of reactive oxygen species (ROS) in plant cells, and high concentrations of ROS can damage plant cells [20]. However, the MAPK pathway plays an essential role in signal transduction of plant cells in response to environmental stress [21]. It has been shown that MKK4/5 in the MAPK pathway was essential for the induction of various defense pathways, including reactive oxygen species (ROS) production, ethylene (ET), and salicylic acid (SA)-mediated defense responses [22]. Transcriptome results showed that low-nitrogen stress first induced changes in abscisic acid signaling, which significantly reduced the expression of MELO3C003408.2 and MELO3C026019.2 in PYR/PYL. Snf1-related protein kinase 2 (SnRK2), a core ABA signaling component, was activated through ABA-triggered type 2C protein phosphatase (PP2C) inhibition and activation, resulting in significant up-regulation of MELO3C020601.2 and MELO3C008882.2 expression in PP2C and SnRK2, causing down-regulated expression of downstream MAPKKK17_18 (MELO3C020535.2), and diminished plant acclimatization to low-nitrogen stress (Figure 5A). Secondly, ER/ERLs sensed low-nitrogen stress, which caused significant down-regulation of the expression of their genes MELO3C007574.2 and MELO3C014002.2, and the cascade of the MAPKK kinase YODA (YDA) and MAPK, which transduced the upstream ligand–receptor signals to the downstream transcription factor SPCH, causing significant MELO3C003421.2 gene down-regulated expression that controlled the stomatal initiation and was directly inhibited by MPK3/6 during phosphorylation (Figure 5B).

Grafting alleviated the low-nitrogen-stress injury to the plants, and the BAK1-related gene MELO3C025664.2 was significantly down-regulated. It caused BAK1 to bind to the LRR receptor serine/threonine protein kinase FLS2, inducing phosphorylation of the related mitogen-activated protein kinases MKK4/5 and MPK1/2. Subsequently, the downstream gene WRKY22 (MELO3C0022675.2 down-regulated) and the related protein FRK1 (MELO3C021017.2 up-regulated) resulted in an early pathogen defense response. Early defense response for the pathogen was expressed through the up-regulation of MELO3C017496.2 and MELO3C018544.2, and late defense response for the pathogen. In addition, pathogen attacks induced phosphorylation of the related mitogen-activated protein kinase MPK3/6 and significant down-regulation of the expression of the Acs6-related gene MELO3C0021182.2, ethylene synthesis (Figure 5C). MEKK1-MKK2 and MEKK1-MKK1 mediate salt and drought and H_2_O_2_-induced activation of MPK4, respectively. MPK4 was a negative regulator of plant immunity [23], and MKS1 interacted with the transcription factor WRKY33, which controlled the regulation of PAD3 and CYP71A13, two genes required for plant resistance (Figure 5C). It has been shown that in Arabidopsis thaliana, the signaling network of MKK4/5 and MPK3/6 in MAPK kinases acts in abiotic and biotic stress responses and regulates embryo and stomatal development [24]. After grafting, the expression of MKK4/5 (significantly up-regulated by MELO3C007574.2, MELO3C014002.2, MELO3C016916.2, and MELO3C026677.2) could be induced, which promoted the stomatal development and strengthened the plant’s ability to cope with low-nitrogen stress (Figure 5D).

### 2.6. Differentially Expressed Genes (DEGs) Involved in Nitrogen Metabolism

Nitrogen is an essential nutrient for plant growth and development, and the metabolic pathways of nitrogen in plants usually include absorption, transport, assimilation, and recycling [25]. In higher plants, the conversion of inorganic nitrogen to organic nitrogen begins primarily through nitrate absorption by root cells, followed by reduction and assimilation to other tissues. Nitrate is the most important form of nitrogen in the soil. The uptake of nitrate by the root system and its transport throughout the plant level involves many transporters and affects plant performance. Four families of nitrate-transporting proteins have been identified so far: nitrate transporter 1/peptide transporter family (NPF), nitrate transporter two families (NRT2), the chloride channel family (CLC), and slow anion channel-associated homologs (SLAC/SLAH) [26]. Three NRT-related genes (MELO3C006395.2, MELO3C024258.2, MELO3C019772.2) (Figure 6A), one CLC-related gene (MELO3C008197.2) (Figure 6B), one SLAC/SLAH-related gene (MELO3C005850.2) (Figure 6C), three NIA(NR) and NII(NIR) (MELO3C022772.2, MELO3C010444.2, MELO3C019505.2) (Figure 6D), and one NLP-related gene (MELO3C017025.2) (Figure 6E) were identified in the transcriptome data. The results showed that low-nitrogen stress affected the expression of the MELO3C019772.2, MELO3C008197.2, MELO3C005850.2, MELO3C022772.2, MELO3C010444.2, and MELO3C019505.2 genes. After grafting, the expression of MELO3C024258.2 and MELO3C017025.2 was significantly up-regulated, alleviating the effect of low-nitrogen stress on their expression (Figure 6).

### 2.7. Differentially Expressed Genes (DEGs) Involved in Carbon Metabolism

Carbon metabolism and nitrogen metabolism are two of the most important physiological processes in plants and are closely related [27]. KEGG enrichment analysis showed that sucrose and starch metabolism pathways were significantly altered during low-nitrogen stress. We analyzed the differentially expressed genes of the “sucrose and starch metabolism” pathway (Figure 7A–K). Under low-nitrogen stress, these genes encode hexokinase, beta-fructofuranosidase, endoglucanase, trehalose 6-phosphate synthase/phosphatase, alpha-trehalase, and beta-amylase. Among the genes encoding by these enzymes, MELO3C014574.2, MELO3C005363.2, MELO3C009488.2, MELO3C024383.2, MELO3C005748.2, MELO3C014430.2, MELO3C015964.2, MELO3C005751.2, MELO3C014105.2, and MELO3C023067.2 were significantly down-regulated. After grafting, the types of encoded enzymes remained almost unchanged, but among the genes encoded by these enzymes, MELO3C003760.2, MELO3C005290.2, MELO3C016287.2, MELO3C007734.2, and MELO3C025477.2 were significantly up-regulated in expression. In summary, low-nitrogen stress may have affected melon self-rooted and grafted seedlings’ carbon and nitrogen metabolic pathways.

### 2.8. Differentially Expressed Genes (DEGs) Involved in Secondary Metabolites

Both nitrogen deficiency and grafting affected secondary metabolic reactions and altered the biosynthesis of secondary metabolites, including phenylpropane [28]. It was involved in resistance to photooxidative stress and played a complementary role as an antioxidant enzyme under severe weather [29]. In this study, we found that the expression of MELO3C011872.2 in the phenylpropane metabolic pathway was down-regulated under low-nitrogen stress. Plant secondary metabolites were usually defense compounds responding to various biotic and abiotic stresses [30]. After grafting, the expression of MELO3C004373.2 and MELO3C022373.2 was significantly up-regulated, thus stimulating the synthesis of secondary metabolites phenylpropane and helping the plants better cope with low-nitrogen stress (Figure 8).

### 2.9. Validation of RNA-Seq Data by qRT-PCR

To verify the DEGs identified by RNA-Seq in relation to the individual signaling pathways, we performed qRT-PCR analysis on samples collected from graft junction tissues of different treatments. Ten selected genes were identified between the two datasets at the expression level (Figure 9). The results showed that the expression of the ten genes detected by qRT-PCR matched the trend of their FPKM value change in RNA-Seq.

## 3. Discussion

Under low-nitrogen stress, most plants undergo significant changes in growth and development. Previous studies have shown that plant acclimatization to nutrient stress in the field mainly depends on morphological changes, and its chlorophyll content is also significantly suppressed [28,31]. In this study, low-nitrogen stress significantly suppressed fluorescence property indices such as quantum yield of non-regulated energy dissipation at Y(NO)PSI, quantum yield of regulated energy dissipation at Y(NPQ)PSII, effective photochemical efficiency of Y(I)PSI, and quantum yield of non-photochemical energy dissipation at PSI induced by the lateral restriction of the Y(NA) receptor. Low-nitrogen stress significantly reduced chlorophyll fluorescence properties in leaves of self-rooted oriental melon seedlings. This was consistent with the findings in rice [32].

The potential transcriptional regulatory mechanisms of cucumber under low-nitrogen stress have been explored by RNA-seq technology [3]. To further reveal the effects of low-nitrogen stress on self-rooted and grafted seedlings of oriental melon, we performed transcriptome analysis. RNA-Seq data showed that low-nitrogen stress mainly induced changes in signaling transduction, oxidative stress, carbon and nitrogen metabolism, and secondary metabolites. Furthermore, we explored the differentially expressed genes involved in hormones, MAPK, sucrose and starch metabolism, and phenylpropane products. Many studies have shown that plant hormones are essential in adaptation to low-nitrogen stress [12]. RNA-Seq data showed that hormone-related genes were mainly enriched in the auxin and brassinolide signaling pathways. Enhanced transport of basal growth hormone from lateral roots outward has been reported under low-nitrogen stress, resulting in lower initial growth hormone content in lateral roots [33], resulting in longer primary root length and reduced root branching. In this study, low-nitrogen stress induced significant changes in AUX/IAA family genes. It was worth noting that MELO3C008287.2 significantly down-regulated expression under low-nitrogen stress. After grafting, it rapidly promoted and benefited cell enlargement and plant growth, as well as improved nitrogen utilization and absorption rate. Under low-nitrogen stress, exogenous brassinolide spray promoted photosynthesis, stimulated stem fructan hydrolysis and spiked sucrose utilization and storage, directed more carbohydrates to the spike, increased the number of fertile wheat flowers, and mitigated wheat floret degradation [34]. Most DEGs involved in brassinolide signal transduction belonged to TCH4 and CYCD3, which regulated the cell elongation and cell division processes. Among them, MELO3C002143.2 of BRI1 in the signal transduction pathway played a significant role in the grafted oriental melon plants, which subsequently caused changes in downstream BSK acting on CYCD3, promoting cell division. In the absence of nitrogen, ROS accumulate excessively in plant cells, causing oxidative stress due to decreased electron frequency in the electron transport system [20,35]. The reactive oxygen species scavenging system played an important role in plant resistance to reactive oxygen species production. The MAPK pathway regulated the production of reactive oxygen species [36]. In our study, low-nitrogen stress caused changes in protein kinases in the MAPK signaling pathway, resulting in significant down-regulation of MAPKKK17_18. After grafting, MELO3C007574.2 and MELO3C014002.2 were significantly induced, which could promote stomatal development. We suggested that MELO3C007574.2 and MELO3C014002.2 genes might improve nitrogen absorption efficiency and enhance low-nitrogen tolerance in grafted oriental melon seedlings. Previous studies have shown that IlWRKY22 can be induced by salt and drought stress and is a key transcription factor regulating flowering time and abiotic stress responses. Transgenic plants overexpressing the WRKY33 gene exhibit enhanced resistance to inundated stress, and WRKY33 was required for plant resistance [37]. However, after grafting, transcription factors WRKY22 and WRKY33 were significantly reduced in this regulatory pathway and did not alleviate the effects of low-nitrogen stress on melon plants.

Grafting has been reported to enhance the absorption of mineral nutrients such as nitrogen, potassium, calcium, and magnesium [38]. In grafted oriental melon plants, CmoNRT2.1 was found to regulate the nitrate absorption capacity of squash rootstock roots [39]. Our study found that the expression of the NRT family-related gene MELO3C024258.2 and NLP family-related gene MELO3C017025.2 were all significantly increased after grafting, which was hypothesized to regulate nitrate absorption and utilization in plants and to enhance low-nitrogen tolerance. Nitrogen metabolism and aromatic amino acid metabolism pathways were enriched in CCRI-69 mainly by regulating starch and sucrose metabolism and carbon metabolism pathways such as glycolysis/gluconeogenesis and pentose phosphate pathways [40]. Further, we analyzed the expression pattern of the “sucrose and starch metabolism” pathway. We found that the expression of many genes encoding enzymes was down-regulated under low-nitrogen stress. After grafting, the types of encoded enzymes remained almost unchanged, while many DEGs (MELO3C003760.2, MELO3C005290.2, MELO3C016287.2, MELO3C007734.2, MELO3C025477.2) were up-regulated. It is speculated that the up-regulated genes identified after grafting may play a key role in coordinating carbon and nitrogen metabolism under low-nitrogen stress. Plants under low-nitrogen stress have reduced photosynthesis and are more sensitive to oxidative stress induced by excess light [41]. As a photooxidative resistor, phenylpropane protects plants from damage under low-nitrogen stress. According to our study, two genes were significantly induced after grafting, and the expression of these genes will facilitate the adaptation of grafted seedlings to low-nitrogen stress.

## 4. Materials and Methods

### 4.1. Plant Materials and Grafting Methods

In this study, the oriental melon ‘HuaBao’ cultivar (*Cucumis melo* var. *makuwa Makino*) and squash ‘ZhenYou 65’ cultivar (*C. moschata*) were used as the scion and rootstock, respectively. Scion seeds were germinated on water filter paper in a dark incubator at 28 °C for 16 h, and rootstock seeds were germinated under the same conditions for 24 h. The germinated seeds were sown in a nutrient soil consisting of peat, vermiculite, and perlite with a volume ratio of 2:1:1. When one leaf of the oriental melon was fully unfolded, it was grafted by apical adhering and a single broken root method [42]. The scion stem was cut diagonally at 45 °C with a sharp blade and a portion of the stem was removed along with the root, and when the rootstock had grown to the point where both cotyledons were fully spread, one cotyledon of the rootstock was removed with a double-sided blade. Then, the two wounds were attached and fixed with appropriately sized clips. Immediately after grafting, the grafted seedling was moved to the grafting room for management, the grafted seedling was gradually supplemented with light, and after 7–9 d of graft healing, the grafted seedlings were moved to a nutrient bowl and then gradually transferred to the greenhouse for management.

### 4.2. Plant Culture and Nitrogen Treatment

In this experiment, a combination of self-rooted and grafted seedlings was used, and two nitrogen treatments were set up: low nitrogen LN (0.4 mmol·L^−1^), normal nitrogen RN (control) (8 mmol·L^−1^), and 9 plants were cultivated in each treatment. The formulation of massive elements is shown in Table 2, and the dosage of micronutrients is shown in Table 3. The experiment was conducted in a 12-hole blue hydroponic growing box with a single trough of 38 cm in length, 28 cm in width, and 14 cm in depth. The nutrient solution was changed every 3 d according to the treatment formulation and the pH of the nutrient solution was measured to maintain pH = 6 or so, and the rest of the plant was the conventional cultivation and management measures. Samples were taken at 30 d of hydroponic culture. All randomly selected robust plants with uniform growth, three replications were set up, and each sample was taken on a sunny day from 9:00 a.m. to 11:00 a.m., and the measured plants were labeled.

### 4.3. Determination of Chlorophyll Fluorescence Characteristic Coefficient

The variable fluorescence FV, the maximum photochemical efficiency FV/Fm (PSII), the potential activity FV /Fo, and the non-photochemical bursting factor (NPQ) of photosystem II were calculated according to the method of Demming-Adams [43]. This included FV = Fm − fo, FV /Fm = (Fm − fo)/Fm, and NPQ = (Fm − Fm’)/Fm’. The values of the five leaves of each seedling were measured, the mean was calculated, and this was repeated for all three seedlings in each group.

### 4.4. Library Preparation for Transcriptome Sequencing

Low-nitrogen (LN)- and normal-nitrogen (RN)-treated melon self-rooted and grafted seedlings with uniform growth after 30 d were selected for RNA-Seq, respectively. Total RNA extraction, library construction, and RNA-Seq were performed by Biomarker Technology Co., (Beijing, China, https://www.biocloud.net/ (accessed on 30 September 2023)). RNA concentration was measured using NanoDrop 2000 (Thermo Fisher Scientific, Waltham, MA, USA), and the detailed protocol was previously described [44]. The specific steps of transcriptome library construction and sequencing are as follows: select magnetic beads with Oligo (dT) to enrich eukaryotic mRNA; add Fragmentation Buffer to randomly interrupt the mRNA; use the mRNA as a template to synthesize the first cDNA strand and second strand, and carry out the cDNA purification; the purified double-stranded cDNA will then undergo end repair, add A tails and connected to sequencing junctions, and then used AMPure XP beads for fragment size selection; finally, the cDNA library was enriched by PCR. PE150 mode sequencing was performed using the Illumina NovaSeq 6000 sequencing platform.

### 4.5. Quality Control, Mapping Reads to the Reference Genome, and Annotation

The equivalent nucleotide sequences were converted from the image data obtained from the Illumina platform. The raw sequencing data were first screened to obtain high-quality sequencing data to ensure the smooth progress of subsequent analysis. To ensure the use of qualified samples for transcriptome sequencing, Baimaike Biological Company utilizes advanced molecular biology equipment to test the purity, concentration, and integrity of extracted Total RNA and strictly monitors it. NanoDrop 2000 spectrophotometer is used to test the purity and concentration of RNA to ensure the quality of transcriptome data.

These sequences were aligned with the melon reference genome (http://cucurbitgenomics.org (accessed on 30 September 2023)). During sequence alignment, a multiple-alignment statistical model was used to take into account the structural features of the sequences. Gene and RNA expression levels were normalized using the FPKM metric, which indicates the number of fragments per kilobase transcript.

InterProScan [45] analyzes the GO [46] Orthology results of new genes using InterPro’s integrated database, and after predicting the amino acid sequences of the new genes, the annotation information of the new genes is obtained by comparing the new gene’s amino acid sequences with the Pfam [47] database using the HMMER [48] software (Version: v3.1b2). Pathway significance enrichment analysis was performed on Pathways in the KEGG database, applying a hypergeometric test to identify Pathways that were significantly enriched in differentially expressed genes compared to the whole genomic background. Pathway significance enrichment enables the identification of the most prominent biochemical metabolic pathways and signaling pathways in which genes are involved, in order to obtain annotated information on the pathways.

### 4.6. Quantification of Gene Expression Levels

Differential expression analysis of two conditions/groups was performed using the DE seq [49]. DE seq provides statistical routines for determining differential expression in digital gene expression data using a model based on the negative binomial distribution [50]. The resulting *p*-values were adjusted using Benjamini and Hochberg’s approach to control the false discovery rate. Genes with an adjusted fold Change ≥ 1.5 and *p*-value < 0.01 found by DE seq were assigned as differentially expressed [51].

### 4.7. Enrichment Analysis of GO Enrichment and KEGG Pathway

We used KOBAS software to test the statistical enrichment of differential expression genes in KEGG pathways. KEGG databases [52] and the KEGG Orthology results of the new genes were obtained. InterProScan [45] used the integrated database of InterPro to analyze the GO Orthology results of new genes [53,54].

### 4.8. Quantitative Real-Time PCR (qRT-PCR)

Some differentially expressed genes were selected to validate the accuracy of RNA-Seq data using qRT-PCR preparation with three biological and technical replicates for each sample conducted as described above. According to the manufacturer’s instructions, the first-strand cDNA synthesis kit was performed using a Prime-ScriptTM II First Strand cDNA synthesis kit (Takara Bio, Dalian, China). The primer sets (Appendix A) for each unigene were designed by Primer Premier 5.0. Pro Taq HS SYBR Green premixed qPCR kit provided by Dalian Ruizhen Biotechnology Company was used for real-time fluorescence quantitative detection. Bole Bio-Rad and CFX96 were used to detect gene expression. Each sample was performed in triplicates, and the reference gene was the melon 18s gene.

### 4.9. Statistical Analysis

All data were the mean of at least three replicates and their standard deviations. All experimental data, including those for correlation analyses, were processed using SPSS statistical software V26.0 and expressed as the mean of three biological replicates of RNA sequence results or the mean ± standard deviation (SD) of three biological replicates. Differences between melon samples were assessed at a significance level of 0.05 by one-way ANOVA tests.

## 5. Conclusions

Low-nitrogen stress had a significant inhibitory effect on the growth of melon seedlings, resulting in a decrease in the fluorescence index of melon seedlings. In this study, transcriptome analyses were performed on self-rooted and grafted seedlings of melon to characterize transcripts under nitrogen deficiency at the seedling stage and grafting to alleviate low-nitrogen stress treatments. Our results showed that grafting improved the low-nitrogen tolerance of melon seedlings through altered signal transduction, oxidative stress, nitrogen metabolism, carbon metabolism, and secondary metabolites pathways. Expression analysis of DEGs confirmed that grafting significantly induced the up-regulated expression of the NRT family gene MELO3C024258.2 and NLP family gene MELO3C017025.2. This study provided new insights into the molecular mechanisms underlying the response of melon-grafted seedlings to mitigate low-nitrogen stress.

## Figures and Tables

**Figure 1 ijms-25-08227-f001:**
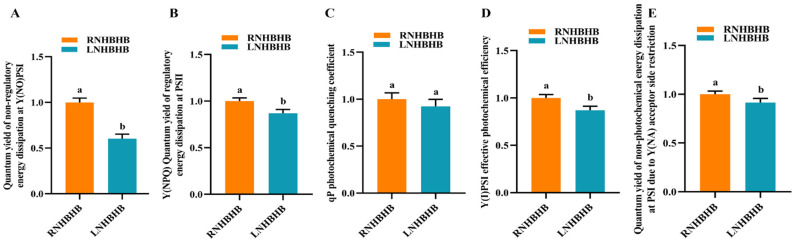
Analysis of fluorescence characterization indexes of self-rooted oriental melon seedlings under low-nitrogen stress: (**A**), the quantum yield of unregulated energy dissipation at Y(NO)PSI. (**B**), the quantum yield of regulatory energy dissipation at Y(NPQ)PSII. (**C**), the photochemical quenching coefficient of qP. (**D**), the effective photochemical efficiency of Y(I)PSI. (**E**), the quantum yield of non-photochemical energy dissipation at PSI induced by lateral restriction of Y(NA) receptors. Different letters indicate significant differences (*p* < 0.05). Values are means ± SD, *n* = 3.

**Figure 2 ijms-25-08227-f002:**
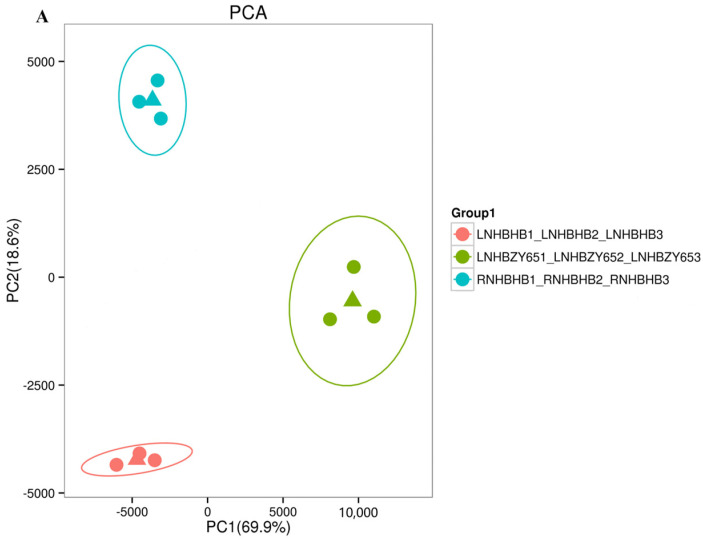
Transcriptome data analysis and functional annotation in the oriental melon leaves under normal nitrogen and low nitrogen treatments: (**A**), the principal component analysis (Circles represent 95% confidence ellipses. The triangle represents the median of the three expressions). (**B**), the related heat map analysis. (**C**,**D**), the analysis of the differentially expressed genes.

**Figure 3 ijms-25-08227-f003:**
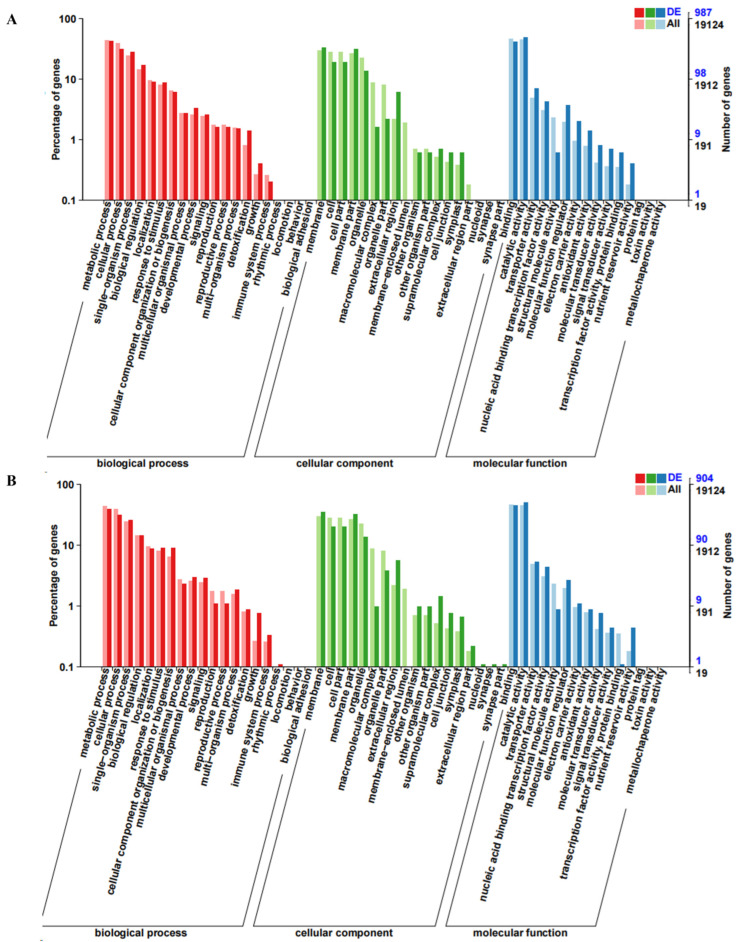
GO and KEGG enrichment analysis of DEGs in the oriental melon leaves under normal nitrogen and low nitrogen treatments: (**A**), GO enrichment analysis of DEGs between RNHBHB and LNHBHB. (**B**), GO enrichment analysis of DEGs between LNHBHB and LNHBZY65. (**C**), KEGG enrichment analysis of DEGs between RNHBHB and LNHBHB. (**D**), KEGG enrichment analysis of DEGs between LNHBHB and LNHBZY65).

**Figure 4 ijms-25-08227-f004:**
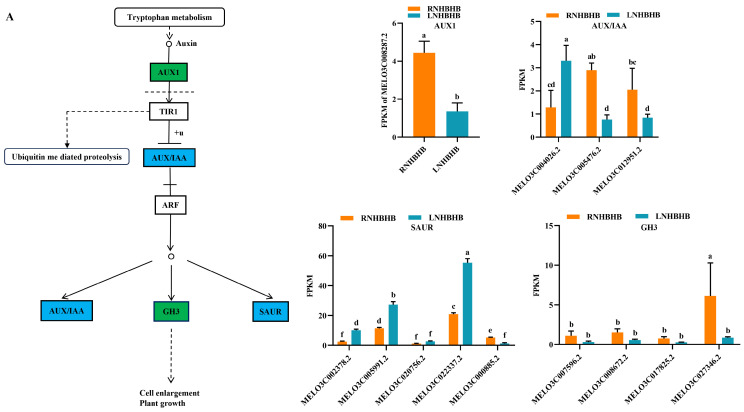
Analysis of DEGs in association with signal transduction pathways: (**A**), Auxin signaling pathway of RNHBHB and LNHBHB; (**B**), Brassinolide signaling pathway of RNHBHB and LNHBHB; (**C**), Auxin signaling pathway of LNHBHB and LNHBZY65; (**D**), Brassinolide signaling pathway of LNHBHB and LNHBZY65. FPKM, fragments read per million maps per thousand base transcripts. Different letters indicate significant differences (*p* < 0.05). Values are means ± SD, *n* = 3.

**Figure 5 ijms-25-08227-f005:**
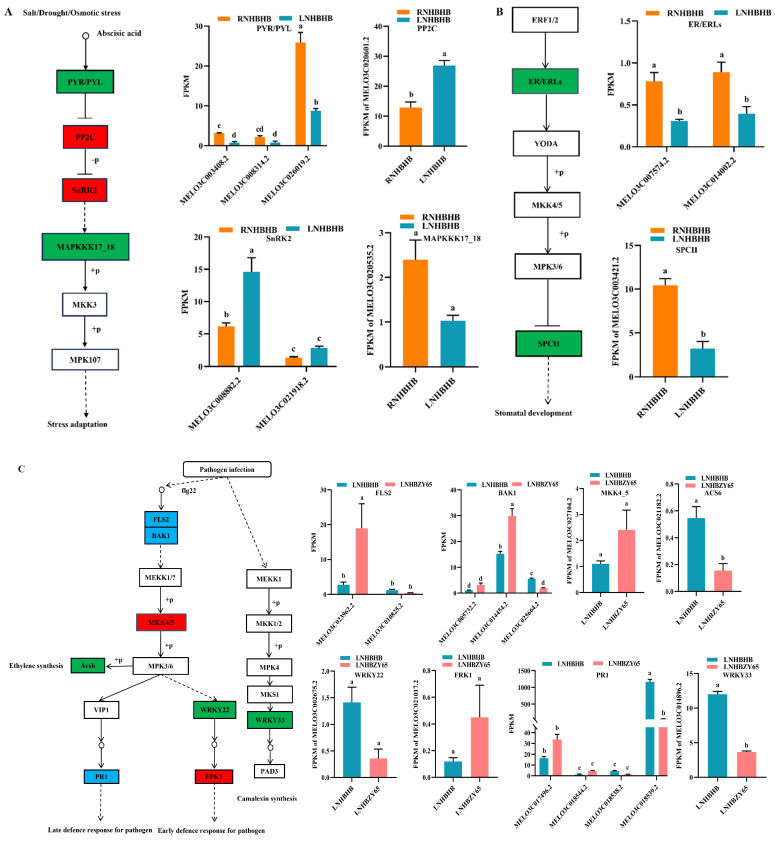
Analysis of DEGs in association with oxidative stress: (**A**), the metabolic pathway of salt/drought/osmotic stress of RNHBHB and LNHBHB; (**B**), the wounding metabolic pathway analysis of DEGs in MAPK signaling pathway of RNHBHB and LNHBHB. (**C**), the metabolic pathway of pathogen infection of LNHBHB and LNHBZY65; (**D**), the wounding metabolic pathway analysis of DEGs in MAPK signaling pathway of LNHBHB and LNHBZY65. FPKM, fragments read per million maps per thousand base transcripts. Different letters indicate significant differences (*p* < 0.05). Values are means ± SD, *n* = 3.

**Figure 6 ijms-25-08227-f006:**
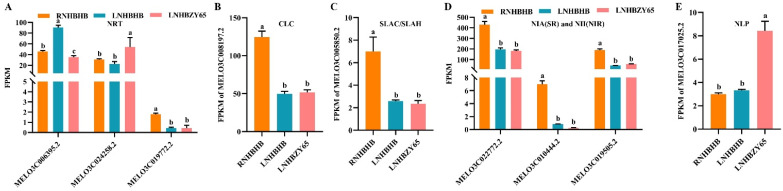
Analysis of DEGs in nitrogen metabolism: (**A**), NRT gene family analysis. (**B**), CLC gene family analysis. (**C**), SLAC/CLAH gene family analysis. (**D**), NIA(SR) and NII(NIR) gene family analysis. (**E**), NLP gene family analysis. FPKM: fragments read per million maps per thousand base transcripts. Different letters indicated significant differences (*p* < 0.05). Values are mean ± SD, *n* = 3.

**Figure 7 ijms-25-08227-f007:**
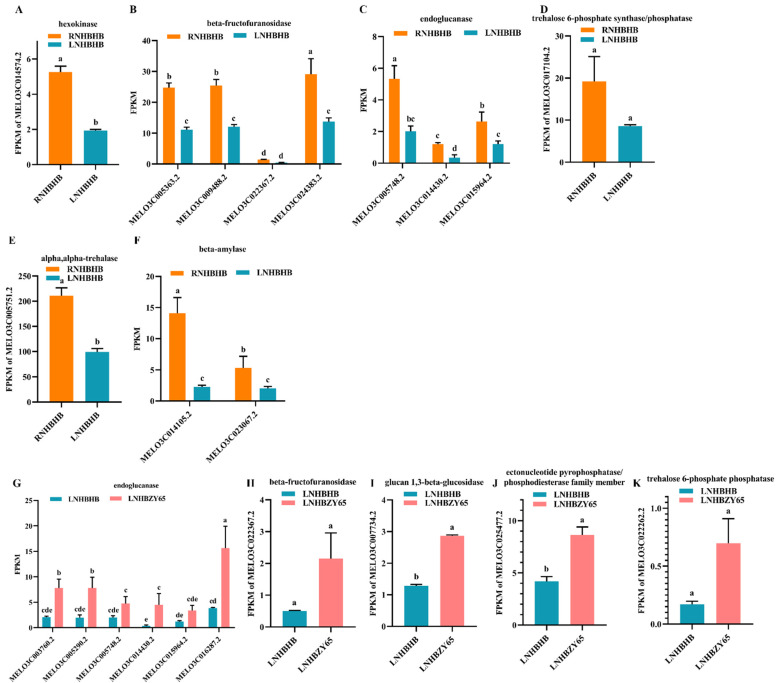
Analysis of DEGs in carbon metabolism: (**A**–**F**), analysis of DEGs in sugar and starch metabolic pathways of RNHBHB and LNHBHB. (**G**–**K**), analysis of DEGs in sugar and starch metabolic pathways of LNHBHB and LNHBZY65. FPKM: fragments read per million maps per thousand base transcripts. Different letters indicated significant differences (*p* < 0.05). Values are mean ± SD, *n* = 3.

**Figure 8 ijms-25-08227-f008:**
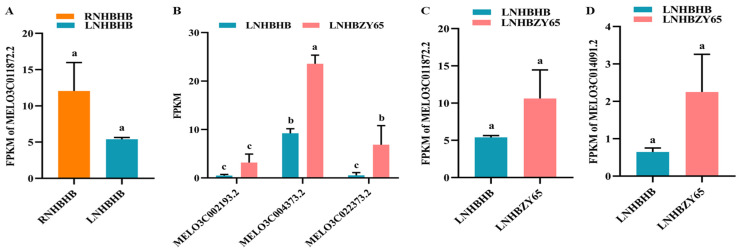
Analysis of DEGs in secondary metabolites: (**A**), analysis of DEGs in secondary metabolites of RNHBHB and LNHBHB; (**B**–**D**), analysis of DEGs in secondary metabolites of LNHBHB and LNHBZY65; FPKM: fragments read per million maps per thousand base transcripts. Different letters indicated significant differences (*p* < 0.05). Values are mean ± SD, *n* = 3.

**Figure 9 ijms-25-08227-f009:**
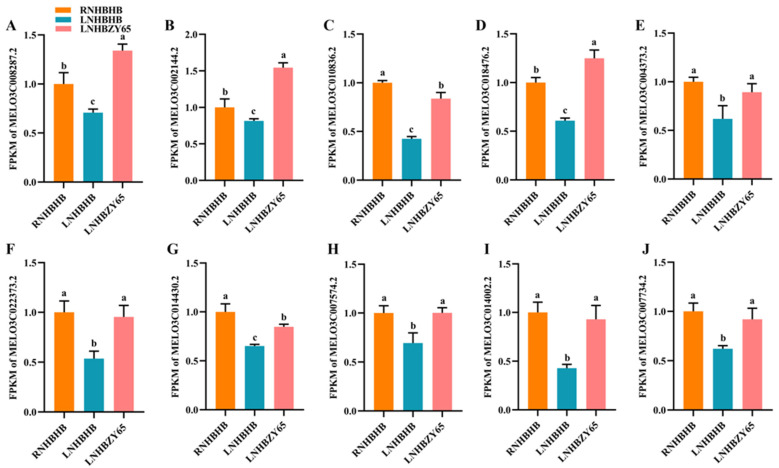
Verification of differentially expressed genes by qRT-PCR: (**A**), FPKM of MELO3C008287.2; (**B**), FPKM of MELO3C002144.2; (**C**), FPKM of MELO3C010836.2; (**D**), FPKM of MELO3C018476.2; (**E**), FPKM of MELO3C004373.2; (**F**), FPKM of MELO3C022373.2; (**G**), FPKM of MELO3C014430.2; (**H**), FPKM of MELO3C007574.2; (**I**), FPKM of MELO3C014002.2; (**J**), FPKM of MELO3C007734.2. Different letters indicate significant differences (*p <* 0.05). Values are means ± SD, *n* = 3.

**Table 1 ijms-25-08227-t001:** Summary of oriental melon transcriptome assemblies.

#Sample ID	Obtained Reads	Obtained Base (bp)	Q20 (%)	Q30 (%)	GC (%)
LNHBHB 1	38697041	11534617118	99.07	94.04	44.89
LNHBHB 2	37251629	11094248198	99.05	93.93	45.00
LNHBHB 3	36228540	10801550260	99.09	94.15	44.85
RNHBHB 1	27329828	8144594918	98.92	93.14	45.04
RNHBHB 2	21551946	6410892762	99.11	94.28	44.94
RNHBHB 3	34739882	10352846224	99.05	93.95	45.10
LNHBZY65 1	32237175	9538969616	99.24	95.04	44.91
LNHBZY65 2	37489331	11158639468	99.05	93.94	45.39
LNHBZY65 3	20393405	6074389982	99.16	94.59	45.01

Note: LNHBHB, the self-rooted oriental melon ‘HuaBao’ with low-nitrogen treatment. RNHBHB, the self-rooted oriental melon ‘HuaBao’ with normal-nitrogen treatment. LNHBZY65, the seedlings of oriental melon scion ‘HuaBao’ grafted onto squash rootstock ‘ZhenYou 65’ with low-nitrogen treatment. Sample ID, project sample name. Obtained Reads, number of reads obtained. Obtained Base (bp), number of bases obtained. Q20 (%), the percentage of bases with a mass value greater than or equal to 20. Q30 (%), the percentage of bases with a mass number greater than or equal to 30. GC (%), GC content of the sample.

**Table 2 ijms-25-08227-t002:** Bulk element formula (unit: mg·L^−1^).

Treatment	Ca(NO_3_)_2_·4H_2_O	KNO_3_	NH_4_H_2_PO_4_	K_2_SO_4_	MgSO_4_·7H_2_O
RN	944.00	0	114.00	522.00	492.00
LN	47.20	0	114.00	522.00	492.00

**Table 3 ijms-25-08227-t003:** Microelement universal nutrient solution formula (unit: mg·L^−1^).

Chemical Name	Grams of Compounds per Liter of Water
EDTA·Na_2_	18.60
H_3_BO_3_	2.86
MnSO_4_·H_2_O	2.13
ZnSO_4_·7H_2_O	0.22
CuSO_4_·5H_2_O	0.08
(NH_4_)6Mo_7_O_24_·4H_2_O	0.02
FeSO_4_·7H_2_O	13.90

## Data Availability

The raw sequencing data are available at NCBI Sequence Read Archive (SRA): PRJNA1129654.

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
