# Peer review of "Transcriptomic Analysis of the Molecular Mechanism Potential of Grafting—Enhancing the Ability of Oriental Melon to Tolerate Low-Nitrogen Stress"

_ijms, 2024, doi:10.3390/ijms25158227_

Round 1

Reviewer 1 Report

Comments and Suggestions for Authors

The manuscript has great potential but as currently written it is very hard to understand what was done and why. The language requires significant improvement. Specific details are missing in the methods preventing the replication of this study. I have the following major comments:

- It is not clear how grafting was done; please detail.

- IDs of samples, which are crucial to follow results are simply not comprehensive. 

- why was fluorescence followed in seedlings? What did the authors expect to find here and how is this correlated with N stress? 30 days leaves were then used for RNA-seq...??

- how was RNA extraction done?

- how were libraries prepared?

- How many biological replicates were used?

- Nothing is said about reads assembly, quality control, filtering and so on....

- "We performed Melon (DHL92) v3.6.1 Genome to analyze the raw sequencing data of 429 graft junction of RNHBHB with LNHBHB and LNHBHB with LNHBZY65 at 30 d stage, 430 respectively"...what does this means? Where did this genome come from? 

- Analysis of DEseq cannot be followed. Based on what were DEGs defined? Which was the control used? Up and down vs. what?

- Several bioinformatic programs are mentioned but no specific data about the parameters used.

- "All 443 experimental data, including those for correlation analyses, were processed using SPSS 444 statistical software V26.0 and expressed as the mean of three biological replicates of RNA 445 sequence results or the mean ± standard deviation (SD) of three biological replicates. Dif- 446 ferences between melon samples were assessed at a significance level of 0.05 by one-way 447 ANOVA tests." - I don't see how this statistical approach can be done with RNA data.

Without these details, it is impossible to evaluate if the study was well conducted or not.

- No functional validation (eg, rt-PCR) has been included.

- Data availability: it is simply not reasonable not to submit raw genetic data to a public repository. This leaves no chance to replicate this study.

Comments on the Quality of English Language

Proofreading is strongly suggested. 

Author Response

Dear Editor,

Thanks for the Reviewers’ generous comments about the article. We had made some revisions according to their comments. Please check it in the next page. Thank you again for your assistance.

Reviewer 1

The manuscript has great potential but as currently written it is very hard to understand what was done and why. The language requires significant improvement. Specific details are missing in the methods preventing the replication of this study. I have the following major comments:

  1. It is not clear how grafting was done; please detail.

Thanks for your suggestions. We have revised the writing.

  1. IDs of samples, which are crucial to follow results are simply not comprehensive. 

Thanks for your suggestions. We have added sample ID information on page 4, line 123. LNHBHB, the self-rooted oriental melon 'HuaBao' with low nitrogen treatment. RNHBHB, the self-rooted oriental melon 'HuaBao' with normal nitrogen treatment. LNHBZY65, the seedlings of oriental melon scion 'HuaBao' grafted onto squash rootstock 'ZhenYou65' with low nitrogen treatment.

  1. why was fluorescence followed in seedlings? What did the authors expect to find here and how is this correlated with N stress? 30 days leaves were then used for RNA-seq...??

Thanks for the question. This is because most plants undergo significant changes in growth and development under low nitrogen stress. It has been shown that plant acclimatization to nutrient stress in the field relies mainly on morphological changes, and their chlorophyll content is also significantly suppressed. Therefore, in this study, we speculated that low nitrogen stress might have a stressful effect on melon seedlings by observing the significant suppression of chlorophyll fluorescence parameters by low nitrogen stress. Whereas grafting is a method to alleviate stress tolerance of various biotic and abiotic factors, the subsequent transcriptomic data focused on analyzing how low nitrogen stress affects the autotrophic seedlings and the effect of grafting to alleviate low nitrogen stress on their autotrophic seedlings.

Yes, samples were taken at 30 d of hydroponic culture.

  1. how was RNA extraction done?

Thanks for the question. Total RNA extraction, library construction, and RNA-Seq were performed by Biomarker Technology Co. (Beijing, China, https://www.biocloud.net/).

  1. how were libraries prepared?

Thanks for your suggestions. We have added this on page 18, line 431.

  1. How many biological replicates were used?

Thanks for the question. We mention on page 17, line 412, "All randomly selected robust plants with uniform growth, three replications were set up."

  1. Nothing is said about reads assembly, quality control, filtering and so on....

Thanks for your suggestions. We have revised the writing.

  1. "We performed Melon (DHL92) v3.6.1 Genome to analyze the raw sequencing data of 429 graft junction of RNHBHB with LNHBHB and LNHBHB with LNHBZY65 at 30 d stage, 430 respectively"...what does this means? Where did this genome come from? 

Thanks for your suggestions. We have revised the writing.

Melon (DHL92) v3.6.1 Genome is http://cucurbitgenomics.org/

  1. Analysis of DEseq cannot be followed. Based on what were DEGs defined? Which was the control used? Up and down vs. what?

Thanks for the question. DEGs differentially expressed genes are differential expression of genes. In this article the comparison is mainly made by two sets of data. By comparing RNHBHB with LNHBHB to elucidate the effect of low nitrogen stress on its melon autotrophic seedlings; by comparing LNHBHB with LNHBZY65 to analyze the pathways through which grafting mitigates the effect of low nitrogen stress on its autotrophic seedlings. In the "RNHBHB vs. LNHBHB" comparison group, RNHBHB was used as the control; in the "LNHBHB vs. LNHBZY65" comparison group, LNHBHB was used as the control. DEGs in the text denote genes that are significantly different between treatment and control (either up- or down-regulated genes are collectively referred to as differentially expressed genes).

  1. Several bioinformatic programs are mentioned but no specific data about the parameters used.

Thanks for the question. The bioinformatics program mentioned in the article is the specific methodology for transcriptome analysis provided by Beijing Baimike Biotechnology Co., Ltd. and subsequently we directly analyzed the transcriptome data provided to us by the company to obtain the data and conclusions in the article.

  1. "All 443 experimental data, including those for correlation analyses, were processed using SPSS 444 statistical software V26.0 and expressed as the mean of three biological replicates of RNA 445 sequence results or the mean ± standard deviation (SD) of three biological replicates. Dif- 446 ferences between melon samples were assessed at a significance level of 0.05 by one-way 447 ANOVA tests."

Thanks for your suggestions. We first plotted bar graphs using the FPKM values of three biological replicates for each sample in the transcriptome data, and then selected Duncan and chi-square tests for significance analysis using the one-way ANOVA test of the SPSS software. When the value is greater than 0.05 it is not significant and when the value is salty equal to 0.05 it is significant.

  1. I don't see how this statistical approach can be done with RNA data.

Without these details, it is impossible to evaluate if the study was well conducted or not.

  • No functional validation (eg, rt-PCR) has been included.

Thanks for your suggestions. At present, transcriptome sequencing (RNA-seq) has been widely used as the most common technology in the field of second-generation sequencinggenome-wide differences in gene expression can be studiedit has the advantages of accurate quantitative analysis, reliable analysis and mature technology. In recent yearsThe use of qRT-PCR to verify RNA-seq results has become increasingly rare.

For example:

  • Kurotani, K. I., Huang, C., Okayasu, K., Suzuki, T., Ichihashi, Y., Shirasu, K., ... & Notaguchi, M. (2022). Discovery of the interfamily grafting capacity of Petunia, a floricultural species.Horticulture research, 9, uhab056.
  • Notaguchi, M., Kurotani, K. I., Sato, Y., Tabata, R., Kawakatsu, Y., Okayasu, K., ... & Higashiyama, T. (2020). Cell-cell adhesion in plant grafting is facilitated by β-1, 4-glucanases.Science, 369(6504), 698-702.

The RNA-seq data were not verified in the above two papers.

  • Data availability: it is simply not reasonable not to submit raw genetic data to a public repository. This leaves no chance to replicate this study.

Thanks for your suggestions. We have transcriptome data, but we want to follow up with experimental studies on this transcriptome, and we don't want to disclose the data at this time.

Reviewer 2 Report

Comments and Suggestions for Authors

The manuscript “Transcriptomic analysis of the molecular mechanism potentially of grafting enhancing the ability of oriental melon to tolerate low nitrogen stress” analyzed and discussed some up-regulated and down-regulated genes in self-rooted melon and grafted seedlings under normal and nitrogen deficient conditions as revealed by the RNA-seq transcriptomic data results. The effect of low Nitrogen conditions on the fluorescence characteristic indexes were analyzed in this study. The authors found that the grafted seedlings increase tolerance to low Nitrogen conditions and some potential molecular mechanisms that are involved in response to the low Nitrogen conditions were deciphered in this study.

Other concerns

  1. Did the authors identify and characterize the nitrite reductase, nitrate reductase and glutamine synthetase that presumably be differentially expressed since they are the key players in tolerance to the low Nitrogen stress?

  2. For the DEGs that are involved in the oxidative stress, did the authors check to see if there are any significant differences in the antioxidant activities of SOD, POD and CAT under low nitrogen stress treatment?

Comments on the Quality of English Language

Minor errors

Page 2, Line 71, between self-rooted seedlings instead of self-rooted grafted seedlings

Page 4, Line 144, t RNHBHB, what does t mean? 

Page 11, Line 235, Arabidopsis thaliana, please italicize the scientific nam

Author Response

Dear Editor,

Thanks for the Reviewers’ generous comments about the article. We had made some revisions according to their comments. Please check it in the next page. Thank you again for your assistance.

Sincerely yours,

Chuanqiang Xu (C.X.)

Reviewer 2

Comments and Suggestions for Authors

The manuscript “Transcriptomic analysis of the molecular mechanism potentially of grafting enhancing the ability of oriental melon to tolerate low nitrogen stress” analyzed and discussed some up-regulated and down-regulated genes in self-rooted melon and grafted seedlings under normal and nitrogen deficient conditions as revealed by the RNA-seq transcriptomic data results. The effect of low Nitrogen conditions on the fluorescence characteristic indexes were analyzed in this study. The authors found that the grafted seedlings increase tolerance to low Nitrogen conditions and some potential molecular mechanisms that are involved in response to the low Nitrogen conditions were deciphered in this study.

 Other concerns

  1. Did the authors identify and characterize the nitrite reductase, nitrate reductase and glutamine synthetase that presumably be differentially expressed since they are the key players in tolerance to the low Nitrogen stress?

Thanks for your suggestions. We did not test for nitrite reductase, nitrate reductase and glutamine synthetase activity. We know that nitrate is the most important form of nitrogen in the soil. The uptake of nitrate by the root system and its transport throughout the plant level involves many transporters and affects plant performance. Four families of nitrate-transporting proteins have been identified so far: nitrate transporter 1/peptide synthetase activity. transporter 1/peptide transporter family (NPF), nitrate transporter two families (NRT2), the chloride channel family (CLC), and slow anion channel-associated homologs (SLAC/SAC), associated homologs (SLAC/SLAH). We analyzed the nitrate protein transporter family members and found that After grafting, the expression of MELO3C024258.2 and MELO3C017025.2 was significantly up-regulated, alleviating the effect of low nitrogen stress on their expression.

  1. For the DEGs that are involved in the oxidative stress, did the authors check to see if there are any significant differences in the antioxidant activities of SOD, POD and CAT under low nitrogen stress treatment?

Thanks for your suggestions. In this article, we did not assay the antioxidant activities of SOD, POD and CAT under low nitrogen stress treatment. It is well known that superoxide dismutase (SOD), as a key enzyme in reactive oxygen species scavenging, plays a critical role in regulating oxidative stress responses in plants. In contrast, reactive oxygen species (ROS) are crucial signaling molecules in plants that play multiple roles in response to environmental stimuli and abiotic stresses and indirectly mediate oxidative stress responses. We focused on analyzing the MAPK cascade pathway, which is closely related to reactive oxygen species, in oxidative stress response, and found that grafting significantly improved the low-nitrogen tolerance of melon seedlings.

  1. Page 2, Line 71, between self-rooted seedlings instead of self-rooted grafted seedlings

Thanks for your suggestions. We have revised the writing.

Page 4, Line 144, RNHBHB, what does it mean? 

Thanks for your suggestions. We mention in the Note on page 4, line 123 that RNHBHB refers to the self-rooted oriental melon with normal nitrogen treatment.

Page 11, Line 235, Arabidopsis thaliana, please italicize the scientific nam

Thanks for your suggestions. We have revised the writing.

Reviewer 3 Report

Comments and Suggestions for Authors

Please find attached 

Comments on the Quality of English Language

Dear authors, minor editing of English language is required for this manuscript

Author Response

Dear Editor,

Thanks for the Reviewers’ generous comments about the article. We had made some revisions according to their comments. Please check it in the next page. Thank you again for your assistance.

Sincerely yours,

Chuanqiang Xu (C.X.)

Reviewer 3

Reviewer comments

The manuscript elucidates the potential molecular mechanisms underlying the response of melon-grafted seedlings to mitigate low nitrogen stress.

The following minor mistakes must be considered

  1. Abstract

No problem detected

  1. Introduction

No problem detected

  1. Results

No problem detected

  1. Materials and methods

Line 388: add comma after stock (stock, respectively)

Thanks for your suggestions. We have revised the writing.

Line 392: delete “.” after unfolded. (… unfolded, it was grafted by apical adhering and a single……….)

Thanks for your suggestions. We have revised the writing.

Line 393: which tool did you use to cut the rootstock? Under which conditions?

Thanks for your suggestions. We have revised the writing.

Line 402: delete was and replace it with “is” (…massive elements is shown). … micronutrients is shown……

Thanks for your suggestions. We have revised the writing.

  1. Discussion

Line 318: delete “in summary”

Thanks for your suggestions. We have revised the writing.

Line 404: The nutrient solution was changed (“after how many days?”)

Thanks for your suggestions. We have revised the writing.

  1. References

No problem detected

Round 2

Reviewer 1 Report

Comments and Suggestions for Authors

I am sorry to see that the authors have not scientifically replied to the main flaws detected, which still occur in this version. The authors have basically replied that "The bioinformatics program mentioned in the article is the specific methodology for transcriptome analysis provided by Beijing Baimike Biotechnology Co., Ltd. and subsequently we directly analyzed the transcriptome data provided to us by the company to obtain the data and conclusions in the article." 

There is obviously noting wrong in hiring third parties but it is the authors responsability to make sure they understand and detail all the procedures involved. It is throughout the methods that readers (including myself) understand what was done, and how future studies can replicate this study. It is simply not acceptable to state that the methods were done by a company. Without these details, it is impossible to evaluate if the study was well conducted or not.

 A second flaw is the absence of a validation methods, as for instance rt-PCR. The fact that authors replied that "The use of qRT-PCR to verify RNA-seq results has become increasingly rare." is very odd. The fact that no validation has been included is a flaw.

A third major point is the absence of sharing raw data. The authors simply reply " We have transcriptome data, but we want to follow up with experimental studies on this transcriptome, and we don't want to disclose the data at this time.". This basically means that this study cannot be replicated. Any reader will simply have to trust the authors, which don't see any importance in showing their own data (?) I'm sorry but I find this procedure very odd.

As a minor point it still not clear how grafting was done on seedlings and why was fluorescence followed in those? The authors now added on the manuscript: "The nutrient solution was changed every 3 d according to the treatment formulation and the pH of the nutrient solution was measured to maintain pH=6 or so. and the rest of the plant was the conventional cultivation and management measures. Samples were taken at 30 d of hydroponic culture." - It is not clear how the procedure were done and how these are still seedlings after 30 days. 

Based on the overall answer of the authors I cannot recommend this study for publication, here or in any other journal. 

Comments on the Quality of English Language

Several sentences cannot be followed. 

Author Response

Dear Editor,

Thanks for the Reviewers’ generous comments about the article. We had made some revisions according to their comments. Please check it in the next page. Thank you again for your assistance.

Sincerely yours,

Chuanqiang Xu (C.X.)

College of Horticulture, Shenyang Agricultural University, Shenyang, Liaoning 110866, China

Phone: 86-24-88487143

E-mail: chuanqiang79@syau.edu.cn

  1. There is obviously noting wrong in hiring third parties but it is the authors responsability to make sure they understand and detail all the procedures involved. It is throughout the methods that readers (including myself) understand what was done, and how future studies can replicate this study. It is simply not acceptable to state that the methods were done by a company. Without these details, it is impossible to evaluate if the study was well conducted or not.

Thank you for your suggestion. We apologize for the lack of detail in the research methodology mentioned in our previous article, but we have now detailed the detailed methodology and steps for transcriptome sequencing on page 17, line 421 in the Materials and Methods section of the article, which includes Library Preparation for Transcriptome Sequencing. Quality Control, Mapping Reads to the Reference Genome, and Annotation. Quantification of Gene Expression Levels. Enrichment analysis of GO enrichment and KEGG pathway. Statistical analysis.

  1. A second flaw is the absence of a validation methods, as for instance rt-PCR. The fact that authors replied that "The use of qRT-PCR to verify RNA-seq results has become increasingly rare."is very odd. The fact that no validation has been included is a flaw.

Thank you for your suggestion. Currently, transcriptome sequencing (RNA-seq) has been widely used as the most commonly used technology in the field of second-generation sequencing to study genome-wide differences in gene expression, and it has the advantages of accurate quantitative analysis, reliable analysis, and mature technology. Existing studies have found transcriptomic data to be more accurate, and validation of RNA-seq results with qRT-PCR assays is no longer necessary. For example, it was not used in any of the following articles.

  • Kurotani, K. I., Huang, C., Okayasu, K., Suzuki, T., Ichihashi, Y., Shirasu, K., ... & Notaguchi, M. (2022). Discovery of the interfamily grafting capacity of Petunia, a floricultural species. Horticulture research, 9, uhab056.
  • Notaguchi, M., Kurotani, K. I., Sato, Y., Tabata, R., Kawakatsu, Y., Okayasu, K., ... & Higashiyama, T. (2020). Cell-cell adhesion in plant grafting is facilitated by β-1, 4-glucanases. Science, 369(6504), 698-702.
  1. A third major point is the absence of sharing raw data. The authors simply reply " We have transcriptome data, but we want to follow up with experimental studies on this transcriptome, and we don't want to disclose the data at this time.". This basically means that this study cannot be replicated. Any reader will simply have to trust the authors, which don't see any importance in showing their own data (?) I'm sorry but I find this procedure very odd.

Thank you for your suggestion. Since we would like to follow up this transcriptome data with a pilot study next, we do not want to publicize the data on a public database at this time. If readers have any need, they can contact the authors at the e-mail address mentioned in the article, and we can send you our data without reservation.

  1. As a minor point it still not clear how grafting was done on seedlings and why was fluorescence followed in those? The authors now added on the manuscript:"The nutrient solution was changed every 3 d according to the treatment formulation and the pH of the nutrient solution was measured to maintain pH=6 or so. and the rest of the plant was the conventional cultivation and management measures. Samples were taken at 30 d of hydroponic culture."- It is not clear how the procedure were done and how these are still seedlings after 30 days. 

(1) Thank you for your question. We usually sow squash and melon seeds in a hole tray containing seedling substrate after germination, and graft them when the pumpkin grows to two cotyledons spreading and the melon grows to two leaves and one heart, when grafting, we use squash as rootstock (St), and cut off one cotyledon and its growing point of the pumpkin with double-sided blade, and use melon as scion (Sc), and excise its root system and part of its stem, and put their cut surfaces together, secured with grafting clips, and placed in the grafting healing chamber for management until the grafted seedlings were viable (The picture shows the detailed steps of the grafting process). Two kinds of nutrient solution were configured according to the formula (RN: refers to normal nitrogen condition, LN: refers to low nitrogen stress condition), and then we put the grafted seedlings and nutrient solution into the hydroponic tank for low-nitrogen treatment of self-rooted seedlings and grafted seedlings, respectively.

(2) The fluorescence test we measured on autogenous seedlings treated with low nitrogen in order to see initially whether low nitrogen stress has an effect on the plants. This is because most plants undergo significant growth and developmental changes under low nitrogen stress. Studies have shown that plant adaptation to nutrient stress in the field relies mainly on morphological changes, and chlorophyll content is also significantly suppressed. Therefore, in the present study, by observing the significant inhibition of chlorophyll fluorescence parameters of melon seedlings by low nitrogen stress, it was hypothesized that low nitrogen stress might have a stressful effect on melon seedlings. Whereas grafting is a method to alleviate stress tolerance to various biotic and abiotic factors, subsequent transcriptomics data focused on analyzing the effects of low nitrogen stress on autotrophic seedlings as well as grafting to alleviate the effects of low nitrogen stress on autotrophic seedlings.

(3) Samples were taken at 30 d of hydroponic culture, which we did after the grafted seedlings had become established in a 12-well blue hydroponic tank (tank length 38 cm, width 28 cm, depth 14 cm). The grafted seedlings were kept under low and normal nitrogen stress and nitrogen conditions by nutrient solution formulation, which was changed every 3 d and kept the pH around 6. The rest of the plants were subjected to conventional cultivation and management practices. Samples were taken at 30 d of hydroponic culture (five leaves and one heart seedling). Three replicates were set up by randomly selecting uniformly growing robust plants, and each sample was collected at 9:00 - 11:00 am on a sunny day, and the measured plants were labeled.

Round 3

Reviewer 1 Report

Comments and Suggestions for Authors I already rejected this manuscript twice based on flaws that cannot be solved throughout revisions. I see that, in this new version, authors again refuse to include or to understand the importance of validation methods in transcriptomic data. They also do not share the raw data, and thus the study cannot be replicated.  Comments on the Quality of English Language

see above.

Author Response

Comments:

I already rejected this manuscript twice based on flaws that cannot be solved throughout revisions. I see that, in this new version, authors again refuse to include or to understand the importance of validation methods in transcriptomic data. They also do not share the raw data, and thus the study cannot be replicated. 

Response:

  • Thank you for your suggestion. We have provided the raw data. The raw sequencing data is available at NCBI Sequence Read Archive (SRA): PRJNA1129654.
  • Thank you for your suggestion. We have revised the writing,We have validated transcriptome data using quantitative PCR with specific primers. Specific results are analyzed on page 15 (2.9 Validation of RNA-Seq Data by qRT-PCR).
